# Medium and Long-Term Outcome of Superselective Transcatheter Arterial Embolization with Lipiodol–Bleomycin Emulsion for Giant Hepatic Hemangiomas: Results in 241 Patients

**DOI:** 10.3390/jcm11164762

**Published:** 2022-08-15

**Authors:** Bing Yuan, Jin-Long Zhang, Feng Duan, Mao-Qiang Wang

**Affiliations:** 1Medical School of Chinese PLA, Beijing 100853, China; 2Department of Radiology, Beijing Tongren Hospital, Capital Medical University, Beijing 100730, China; 3Departments of Interventional Radiology, Chinese PLA General Hospital, Beijing 100853, China

**Keywords:** angiography, bleomycin, embolization, therapeutic, liver hemangioma

## Abstract

Purpose: To evaluate the medium and long-term efficacy of superselective transcatheter arterial embolization (TAE) with lipiodol–bleomycin emulsions (LBE) for giant hepatic hemangiomas. Methods: A total of 241 patients who had underwent TAE with LBE for hepatic hemangiomas from January 2010 to December 2016 were retrospectively reviewed. Blood tests were performed 3 and 7 days after TAE and procedural-related complications were recorded. The patients were followed up by enhanced CT or MRI imaging at 6, 12, 36, and 60 months post-TAE, respectively. Technical success of TAE was defined as successful embolization of all identifiable arteries supplying to the hemangiomas. Clinical success was defined as improvement of the abdominal symptoms and indications on the imaging examinations that the hemangiomas had decreased by more than 50% in maximum diameter. Results: TAE was performed successfully in all patients without serious complications. Improvement of the abdominal symptoms was recorded in 102/102 cases (100%). The reduction rate of the tumor maximum diameter with >50% at 6, 12, 36, and 60 months was 88.1% (190/210), 86.7% (170/196), 85.2% (124/142), and 86.5% (45/52), respectively. There was a significant change from pre-TAE to follow-up values in maximum diameter (*p* < 0.05). Conclusion: TAE with LBE was feasible and effective for giant hepatic hemangiomas. The reductions of the tumor maximum diameter with >50% at medium (≥3 years) and long-term (≥5 years) follow-up were satisfactory, with 85.2% and 86.5%, respectively.

## 1. Introduction

Hepatic hemangioma is one of the most common benign hepatic tumors, with an incidence of approximately 0.4–20.0% [1,2]. Cavernous hemangiomas constitute the most common pathological type of hepatic hemangioma. They are also more prevalent in women. In recent years, with advancements in imaging technology, the detection rate of hepatic hemangiomas has increased significantly. Traditionally, the preferred treatment modality was surgical resection [3], but for patients with multiple lesions or lesions adjacent to the hepatic portal vessels, surgical resection often has a suboptimal outcome, and is associated with a high risk of bleeding [4]. Superselective transcatheter arterial embolization (TAE) is, therefore, now widely used as an effective treatment modality for hepatic hemangiomas. However, previous reports have indicated that the treatment efficacy and adverse events (AEs) are diverse [5,6]. The objective of the present study was to retrospectively analyze the feasibility and medium to long-term efficacy of lipiodol–bleomycin emulsions (LBE) in TAE for giant hepatic hemangiomas.

## 2. Materials and Methods

### 2.1. Ethics Statement

This retrospective single-center study was approved by the institutional review board (no further approval number due to it being a retrospective study), and was performed in accordance with the ethical standards laid out in the 1964 Declaration of Helsinki and its later amendments. All patients provided written informed consent to undergo the procedure and contribute their clinical and imaging outcome data to the study.

A total of 241 patients who had undergone TAE of giant hepatic hemangiomas from January 2010 to December 2016 were evaluated. The inclusion criteria were as follows: (1) hepatic hemangiomas with a tumor diameter of >5 cm, with progressive increase in size; (2) upper abdominal discomfort or pain typically caused by the hemangiomas; (3) unsuitable for resection as evaluated by two hepatobiliary surgeons or the patients refused surgery. The exclusion criteria were as follows: (1) allergic to iodine-containing contrast agents; (2) severe liver and renal insufficiency; (3) unregulated coagulation parameters; (4) abdominal symptoms excluding other hepatobiliary or gastrointestinal disorders.

All patients were diagnosed with enhanced liver computed tomography (CT) or magnetic resonance imaging (MRI). The images were evaluated for the size, number, and location of the hemangiomas. Technical success of TAE was defined as successful embolization of all identifiable arteries supplying the hemangiomas. Clinical success was defined as improvement of the abdominal symptoms and indications on the imaging examinations that the hemangiomas had decreased by more than 50% in maximum diameter. The abdominal symptoms were assessed by physicians from the department of hepatobiliary surgery prior to TAE; the abdominal pain was graded as no pain (0), light pain (1–2), moderate pain (3–5), severe pain (6–7), and very severe pain (8–10). 

TAE was performed by an interventional physician with 30 years of experience in vascular interventional radiology, using a digital flat-panel detector system (GE Innova 4100-IQ large flat-panel angiographic imaging system, GE Healthcare, Milwaukee, WI, USA) and nonionic contrast medium (Ultravist, 370 mgI/mL; Bayer healthcare Co., Ltd. Guangzhou Branch, Guangzhou, China). The right femoral artery was punctured after local anesthesia, and a 4 Fr vascular sheath (Terumo, Tokyo, Japan) was placed using Seldinger’s technique. Celiac artery-hepatic artery and superior mesenteric artery angiography (indirect portal angiography) were all carried out with a 4 Fr hepatic artery catheter (Cordis, Miami, FL, USA). Inferior phrenic artery angiography was performed as appropriate for lesions adjacent to the diaphragm. Afterwards, a 2.6 Fr microcatheter (Asahi Intecc Co., Tokyo, Japan) was inserted into the feeding arteries of the hemangiomas, and the LBE was slowly injected under the guidance of X-ray fluoroscopy. The emulsion, composed of 15 IU bleomycin powder (bleomycin hydrochloride for injection, Zhejiang Haizheng Pharmaceutical), was dissolved in 3 mL of normal saline, and then mixed with 10–15 mL of Lipiodol (Guerbet, Villepinte, France) with a concentration of 1.5 mg bleomycin/1 mL lipiodol. The maximum dose of bleomycin was 15 IU/session; the dose of lipiodol (mL) varied depending on the size of the hemangiomas, but was limited to 20 mL/session. The emulsion was mixed with a three-way stopcock. After embolization, control hepatic arteriography was performed to confirm the complete occlusion of the target arteries. The end point of TAE was considered when all arterial branches that supplied the hemangiomas were completely occluded with no observed tumor stain. 

After TAE, a broad-spectrum antibiotic was administered to patients for 2–3 days to prevent infection. Appropriate hydration therapy was delivered to patients who had received a larger dose of contrast agent (>200 mL). Antipyretics, analgesics, and other medications were administered for symptomatic management as needed. Blood tests (blood routine, liver function, kidney function, and coagulation function) were performed three and seven days after the procedure and TAE-related complications (fever, pain) were recorded. 

The patients were followed up by improvement of the abdominal symptoms and liver enhanced CT or MRI imaging at 6, 12, 36, and 60 months after TAE. 

### 2.2. Statistical Analysis

Statistical analysis was performed using SPSS software (version 25; SPSS, Armonk, NY, USA). The study’s quantitative variables were expressed as mean values, standard deviations, and minimum and maximum values. Qualitative variables were expressed as numbers and percentages. A paired t-test was used to compare the differences between the pre-TAE values and post-TAE values, One-factor repeated-measures analysis of variance with the use of Bonferroni adjustment was conducted for comparisons against the hemangioma diameters prior to treatment. *p* < 0.05 was considered statistically significant. 

## 3. Results

A total of 241 patients with a mean age of 47 ± 9 years (a range of 27–75 years) were included. There were 139 patients without any discomfort or pain, 52 patients with upper abdominal discomfort, 39 patients with light upper abdominal pain, and 11 patients with moderate upper abdominal pain. The mean diameter of the tumors was 9.5 ± 3.1 cm with a range of 5.1–20 cm (Table 1).

TAE was successfully performed in all patients (100%). All patients with abdominal symptoms were significantly relieved and no patient presented symptomatic recurrence during the follow-up period. The improvement of the abdominal symptoms at 6 and 12 months after TAE were 91/102 (91%) and 9/102 (9%), respectively. The reduction rate of the tumor maximum diameter with >50% at 6, 12, 36, and 60 months was 88.1% (190/210), 86.7% (170/196), 85.2% (124/142), and 86.5% (45/52), respectively (Figure 1 and Figure 2). At 6, 12, 36, and 60 months after TAE, the hepatic hemangiomas diameters were 4.6 ± 1.4 cm (range, 1.1–10.5 cm), 3.0 ± 1.3 cm (range, 0.6–10.0 cm), 3.0 ± 1.2 cm (range, 0–9.1 cm), and 2.9 ± 1.2 cm (range, 0-8.8 cm), respectively. There was a significant reduction from pre-TAE to follow-up values in maximum diameter (*p* < 0.05). (Table 2) The shrinkage period was determined to have ended after 12 months. (Figure 3) There was a significant interaction between procedure and time (*p* ≤ 0.05). 

No serious complications were observed after TAE, such as biloma and liver abscess, etc. Transient increases of hepatic enzyme (Aspartate transaminase and Alanine transaminase) and plasma bilirubin values were found in all patients after TAE and returned to normal in 1–2 weeks. Thirty patients (12.4%) experienced minor fever and fifty six patients (23.2%) had minor-to-moderate pain within 3–5 days. The median follow-up period was 33 ± 18 months (range 6–60 months). The cases lost to follow-up after TAE at 6, 12, 36, and 60 months were 31, 45, 99, and 189, respectively.

## 4. Discussion

In the present study, good responses were achieved in the improvement of the abdominal symptoms and reduction of hemangioma diameters after TAE with BLE. The medium (≥3 years) and long-term (≥5 years) clinical success rates were 85.2% and 86.5%, respectively. Furthermore, we found that the greatest decrease in hemangioma size occurred at 6–12 months post-TAE, and almost no changes were observed during the follow-up period thereafter. Based on these findings, the second TAE may be spared when the size reduction of the hemangiomas is unsatisfactory within 6 months after TAE. 

Hepatic hemangiomas are the most common benign liver lesions that are found incidentally on routine abdominal cross-sectional imaging and are usually asymptomatic and remain stable in size, subsequently requiring no treatment [7,8,9]. When patients present with large lesions (>5 cm in diameter) with a progressive increase in size and are symptomatic as related to the lesions, a specific treatment is usually needed [10]. Surgical resection has been the traditional treatment of choice in symptomatic patients or in those with marked enlargement [11,12]. However, surgery has relatively higher complications (bleeding, high cost, and infection), particularly when the lesion exceeds 10 cm in diameter [5]. Furthermore, surgical treatment is not always possible because of the large size or unfavorable location of a lesion, or other patient factors. According to previous studies, percutaneous treatment techniques have become another option for the treatment of hepatic hemangiomas, such as microwave ablation, radiofrequency ablation (RFA), and percutaneous sclerotherapy with bleomycin and ethiodized oil [13,14,15]. However, these techniques are only suitable for tumors less than 7 cm in diameter. Of note, to support the effectiveness of these techniques, more trials have to be performed [6]. In addition, the rates of the complications (hemolysis-related complications and the systemic inflammatory response syndrome) after RFA were high, with 34%–100%, particularly when the tumor was larger than 10 cm in diameter [16,17]. 

Recently, TAE has been identified as a suitable technique for the treatment of hepatic hemangiomas. However, there is a lack of consensus regarding whether TAE is effective in the treatment of hemangiomas and regarding the severity of complications [5,18,19,20,21,22]. Liu et al. [20] reported that TAE is not satisfactory for liver hemangiomas, and carries a risk of severe complications. Four patients (4/55, 7.3%) experienced severe complications, including biloma and abscess due to necrosis of the bile duct. Only 19 patients (19/53, 35.8%) had smaller or similar-sized hemangiomas, possibly due to the presence of multiple nourishing vessels. An opposing conclusion was presented by Torkian et al. [5], who conducted a systematic review and meta-analysis suggesting that TAE with bleomycin, pingyangmycin, or ethanol, in combination with lipiodol, was safe and effective. No mortality was reported, and CIRSE grade 3 complications were reported in 6/1450 cases (0.4%). Clinical response to TAE was reported to be 63.3–100%. In the present study, TAE was performed successfully in all participants without serious complications. In patients with large hemangiomas, most feared complications after TAE such as an abscess in the ischaemic or necrotic tissue. Therefore, a strict aseptic approach is essential, and prolonged use of antibiotics as appropriate is very important. None of the patients developed significant liver function damage post-TAE. This may be due to the absence of chronic hepatic disease in the patients and the non-retention of the embolic agents in normal liver tissue due to super-selective embolization. Caution is necessary, however, not to compromise branches of the normal liver arteries to avoid unnecessary liver damage. All patients with abdominal symptoms were significantly relieved and the hemangiomas diameters significantly reduced from pre-TAE to follow-up in the present study. LBE were selected as the embolization materials and used to embolize the collateral vessels during the TAE. Collateral arteries, such as the inferior phrenic and gastroepiploic arteries, are often involved in feeding hemangiomas. Therefore, the embolization of these collateral vessels is crucial. On the other hand, bleomycin has been widely used for the treatment of benign tumors because of its non-specific inhibition and destruction of vascular endothelial cells [23,24,25]. Dose-related pulmonary fibrosis was reported in some oncology patients receiving high cumulative doses (>400 mg) of intravenous bleomycin. Furthermore, the maximum administered dose of bleomycin (15 IU/per session) in the present study was much lower than the toxic dose. While, a part of patients experienced a partial response of less than 50% reduction in hemangiomas sizes may related to the concentration of bleomycin (reduced in the treatment of excessively large hemangiomas). For this reason, further studies are warranted to investigate the correlation between bleomycin dose and clinical success. Additionally, reopening of collateral vessels and revascularization after TAE may be associated with clinical failure. In the present study, the number of patients who were lost to follow-up increased with time, especially at 36 and 60 months post-TAE. As hepatic hemangiomas are benign hepatic tumors, some patients were monitored with ultrasonography alone once their treatment had been evaluated as a clinical success. We excluded this group of patients because ultrasound is not accurate. Therefore, the loss to follow-up in these patients did not affect the final results of the study.

Our study has several limitations. First, this was a retrospective and single-center study; further prospective studies with multiple centers are necessary. Second, there was a lack of comparison between surgical intervention and other treatments. Third, we did not compare with other types of embolic materials, such as particles (polyvinyl alcohol particles, Embosphere, trisacryl gelatin microspheres, gelatie sponge particles, ethanol, and coils). Finally, there was a relatively high proportion of lost follow-up patients due to the retrospective study nature.

## 5. Conclusions

TAE with BLE for giant hepatic hemangiomas is a safe and effective method of improving abdominal symptoms with reliable medium (≥3 years) and long-term (≥5 years) clinical and radiological success rates.

## Figures and Tables

**Figure 1 jcm-11-04762-f001:**
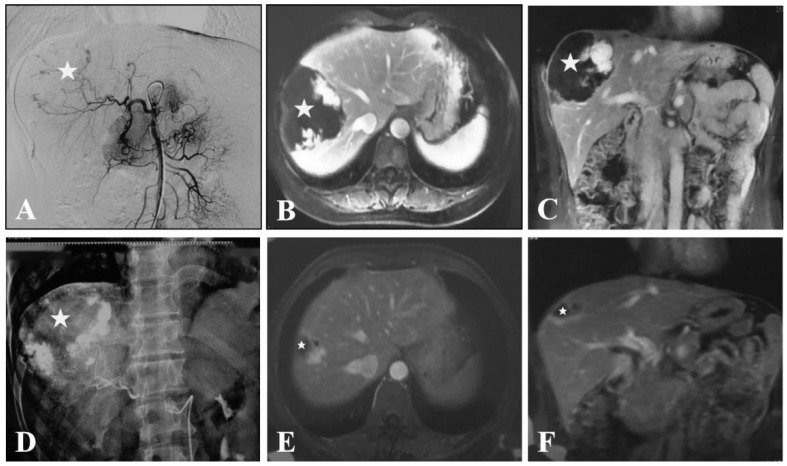
Representative case of significantly reduced tumor with transarterial arterial embolization (TAE) using Lipiodol–Bleomycin emulsion (LBE). A 41-year-old woman presented with right upper quadrant discomfort and the diagnosis of the hemangiomas was made using magnetic resonance imaging (MRI). She was followed up for 5 years after TAE. (**A**). Digital subtraction angiography (DSA) of the hepatic artery, arising from the supper mesentric artery, showed a large hypervascular mass (★) in the right lobe of the liver. (**B**). Axial contrast-enhanced T1-weighted MRI obtained before TAE showed a large hypervascular lesion in the right lobe of the liver (★). (**C**). Coronal contrast-enhanced T1-weighted MRI obtained before TAE showed a large hypervascular lesion in the right lobe of the liver (★). (**D**). Upper abdominal radiogram obtained immediately post-TAE with LBE demonstrated the high dense lipiodol-emulsion retention in the lesion (★). (**E**). Axial contrast-enhanced T1-weighted MRI obtained at 5 years post-TAE showed a small residual lesion (★). (**F**). Coronal contrast-enhanced T1-weighted MRI obtained at 5 years post-TAE showed a small residual lesion (★) with a 95% reduction of the hemangioma. This patient had complete resolution of her abdominal symptoms at 3 months after TAE.

**Figure 2 jcm-11-04762-f002:**
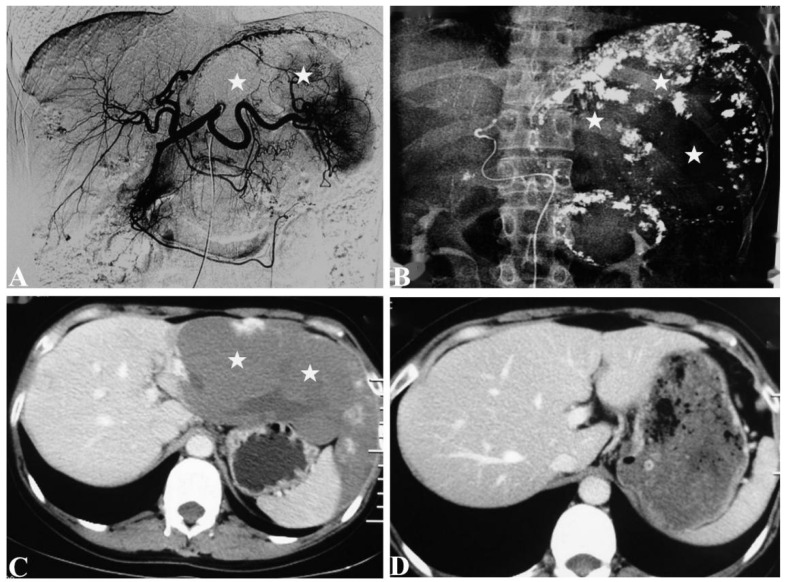
Representative case of complete disappearance of the tumor with TAE using a lipiodol–bleomycin emulsion (LBE). A 31-year-old woman presented with the left upper quadrant discomfort for 3 years. She was followed up for 6 years after TAE. (**A**). Digital subtraction angiography (DSA) of the hepatic artery showed a large hypervascular mass in the left lobe of the liver (★). (**B**). Upper abdominal radiogram obtained immediately post-TAE with LBE demonstrated the high dense lipiodol-emulsion retention in the lesion (★). (**C**). Axial contrast-enhanced computed tomography (CT) at portal venous phase obtained before TAE showed a huge lesion in the left lobes of the liver (★). (**D**). Axial contrast-enhanced CT at portal venous phase obtained at 5 years post-TAE showed complete response of the lesions. This patient had complete resolution of her abdominal symptoms at 6 months after TAE and the hemangioma reduction was 100% at the last follow-up, without recurrence.

**Figure 3 jcm-11-04762-f003:**
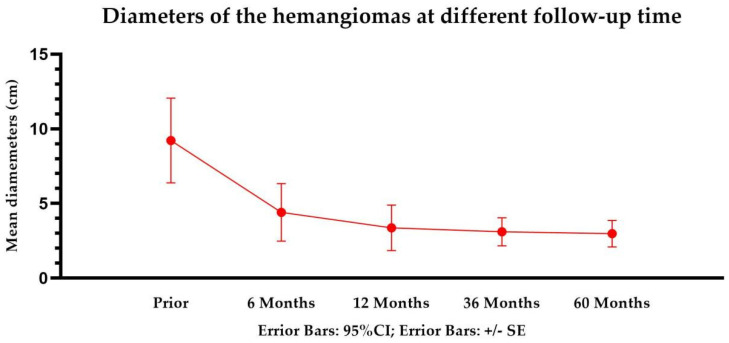
Diameters of the hemangiomas at different follow−up time points.

**Table 1 jcm-11-04762-t001:** Patients’ basic characteristics.

Variables	Baseline	n	Percent (%)
Sex	Male	52	21.6
	Female	189	78.4
Age	<50 years	156	64.7
	≥50 years	85	35.3
Number	Single	42	17.4
	Multiple	199	82.6
Location	Left lobe	19	7.9
	Right lobe	71	29.5
	lobes	151	62.6
Abdominal symptoms	None	139	57.7
	Upper abdominal discomfort	52	21.6
	light pain	39	16.2
	moderate pain	11	4.5
Pre-TAE diameter	<10 cm	178	73.9
	≥10 cm	63	26.1

TAE: transarterial arterial embolization.

**Table 2 jcm-11-04762-t002:** Comparison of the maximum diameters of the hepatic hemangiomas post-TAE during the follow-up period.

Diameter (cm)	n	$\bar{x}$ ± s	Ranges	t	*p*
Pre-TAE	241	9.5 ± 3.1	5.1–20.0	—	—
6 months	210	4.6 ± 1.4	1.1–10.5	30.3	<0.01
12 months	196	3.0 ± 1.3	0.6–10.0	22.6	<0.01
36 months	142	3.0 ± 1.2	0–9.1	15.2	<0.01
60 months	52	2.9 ± 1.2	0–8.8	8.3	<0.01

Values are presented as means ± standard deviations (SD) and ranges; TAE: transarterial arterial embolization.

## Data Availability

Data is contained within the article or request from the corresponding author.

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
