# Peer review of "Medium and Long-Term Outcome of Superselective Transcatheter Arterial Embolization with Lipiodol–Bleomycin Emulsion for Giant Hepatic Hemangiomas: Results in 241 Patients"

_jcm, 2022, doi:10.3390/jcm11164762_

Round 1

Reviewer 1 Report

Dear Authors.

The manuscript was nicely corrected with in relation to  my previous remarks. Thank YOU.

Author Response

Response: Thank you very much for your recognition. The manuscript has benefited from your insightful suggestions.

Reviewer 2 Report

Dear authors, 

There has been a lot of changes to your manuscript mandated by me or other reviewers. In general the clarity of the article improved, but still there are some small stylistic mistakes.

Page 7, line 276, rephrase like this: 

However, these techniques are suitable for tumors less than 7 cm in diameter. Of note, to support the effectiveness of these techniques more trials have to be performer.

Page 8, line 308

In patients with large hemangiomas most feared complication after TAE is an abscess in the ischaemic or necrotic tissue. 

Page 8, line 315

…,- to a avoid ennecessary liver damage

The conclusion si a non-sense at a time, you have to rephrase:

TAE with BLE for giant hepatic hemangiomas is a safe and effective method of improving abdominal symptoms with reliable medium (≥3 years) and long-term (≥5 years) clinical and radiological success rates.

Author Response

Response:

Thank you very much for your recognition and suggestion. The manuscript has benefited from your insightful suggestions. 

We have modified all of these points in the manuscript.

Reviewer 3 Report

I have no further comments or suggestions 

Author Response

Response:

Thank you very much for your recognition. The manuscript has benefited from your insightful suggestions.

This manuscript is a resubmission of an earlier submission. The following is a list of the peer review reports and author responses from that submission.

Round 1

Reviewer 1 Report

This article presents transcatheter arterial embolization technique with bleomycin. Authors discuss its potential value in the treatment of hepatic hemangioma. The aim of the study is acceptable. Although the TAE is nowadays not a standard clinical technique during treatment, presented data support that the TAE with bleomycin provides good early results and may potentially be of value in the treatment process. However, the study has several serious limitations: 1. The number of patients included in the retrospective analysis is 241, and ONLY 52 patients reached the 60 month F-U, and 142 were assessed 36 month after the procedure. If the Authors aimed to asses the mid-term and long-term results of the technique the follow-up should be much better. 2. Significant number of patients with multiple (more than 1) hepatic lesion (82.6%) were included in the study group. In my opinion this could have an impact on the results of TAE. Do the Authors consider to embolize all lesion during only one session? Is the technical success rate related to all treated lesions or the largest one?

Author Response

  1. The number of patients included in the retrospective analysis is 241, and ONLY 52 patients reached the 60 month F-U, and 142 were assessed 36 month after the procedure. If the Authors aimed to asses the mid-term and long-term results of the technique the follow-up should be much better.

Answer: Thank you very much for your suggestion. As hepatic hemangiomas are benign hepatic tumors, some patients were content with ultrasonography alone on review, once their treatment had been evaluated as a clinical success. We excluded this group of patients from the study as ultrasound is not accurate. Therefore, in our view, the loss of follow-up on the part of these patients did not affect the final results in our view.

  1. Significant number of patients with multiple (more than 1) hepatic lesion (82.6%) were included in the study group. In my opinion this could have an impact on the results of TAE. Do the Authors consider to embolize all lesion during only one session? Is the technical success rate related to all treated lesions or the largest one?

Answer: Thank you very much for your suggestion. We embolized all lesions during the TAE procedure, but focusing on the largest one. Clinical efficacy was defined by indications on the imaging examinations that the tumors had decreased by more than 50% in maximum diameter. Therefore TAE was considered a technical success if the bleomycin-lipiodol emulsions were delivered into the largest one.

Reviewer 2 Report

Review of manuscript JCM Nr.1766696

I would like to congratulate the authors on the results of this large retrospective analysis of transcather therapy of large giant cell hemangiomas. In general the article is well written, the results are promising and clear, but there are several issues especially in the discussion section that have to be thorougly  addressed:

1.     Abstract: concise and clear

2.     Page 3, line 104 ad „all“ to Although almost „all“ patients… 

3.     Discussion section should begin with: 

The main findings of our retrospective analysis can be summarized as follows: 1 … 2 … 

4.     Then I would suggest subdivision to : 4.1 Previous studies 4.2 Current study 4.3 Implications for clinical practice (based mainly on your current study) 4.4 Limitations, since at the present time the discussion section is really hard to read and poorly arranged

5.     Page 3, line 130, reorganize, there was barely no change not the ther way round. The Sentece beginning has to be rearranged. Maybe:  Based on our experience, for too large hemangiomas (size in cm) bleomycin may not cover the whole tumor.  

6.     Page 3, line 131, the sentence again does not make sense. You wanted to say, that : Nineteen patients (7.9%) experienced a less than 50% hemangioma reduction (regression is more or less statistical function, do not use it in this context), what may be explained by …. ?

7.     Please comment on the difference between your succes rate (complications)  and others (i.e.Liu X et al.Medicine (Baltimore). 2017 Dec; 96(49): e9029.)

8.     From my point of view the transcather method si not numerically compared to the succes rate of surgery vs. its complications. You also may speculate wether this trnacatheter method may replace surgery or what are possible anatomicaly more feasible variants for surgery and transcatheter approach

9.     Please mention numerically the loss to follow-up

10.  Conclusion should be rephrased: Furthermore the medium and long-term clinical success rates demonstrate efficacy and safety of this interventional approach. (I do not think that longer follow-up would change anything. This statement is not supported by your data)

Author Response

  1. Abstract: concise and clear

Answer: Thank you very much for your recognition.  

  1. Page 3, line 104 ad „all“ to Although almost „all“ patients…

Answer: Thank you very much for your suggestion. We have modified this point in the Results.

  1. Discussion section should begin with: The main findings of our retrospective analysis can be summarized as follows: 1 … 2 …

Answer: Thank you very much for your suggestion. We have modified this point and re-write in the discussion.

  1. Then I would suggest subdivision to : 4.1 Previous studies 4.2 Current study 4.3 Implications for clinical practice (based mainly on your current study) 4.4 Limitations, since at the present time the discussion section is really hard to read and poorly arranged

Answer: Thank you very much for your suggestion. We have modified this point and re-write in the discussion.

  1. Page 3, line 130, reorganize, there was barely no change not the ther way round. The Sentece beginning has to be rearranged. Maybe:  Based on our experience, for too large hemangiomas (size in cm) bleomycin may not cover the whole tumor.  

Answer: Thank you very much for your suggestion. We have modified this point and re-write in the discussion.

  1. Page 3, line 131, the sentence again does not make sense. You wanted to say, that : Nineteen patients (7.9%) experienced a less than 50% hemangioma reduction (regression is more or less statistical function, do not use it in this context), what may be explained by …. ?

Answer: Thank you very much for your suggestion. This may be explained by nineteen patients experienced a partial response with a less than 50% reduction of hemangioma.

  1. Please comment on the difference between your succes rate (complications)  and others (i.e.Liu X et al.Medicine (Baltimore). 2017 Dec; 96(49): e9029.)

Answer: Thank you very much for your suggestion. We have modified this point in the discussion. Superselective embolization and prolonged use of antibiotics may reduce the complications. And the complete occlusion of the trophoblastic vessels may improve the success rate.   

  1. From my point of view the transcather method si not numerically compared to the succes rate of surgery vs. its complications. You also may speculate wether this trnacatheter method may replace surgery or what are possible anatomicaly more feasible variants for surgery and transcatheter approach

Answer: Thank you very much for your suggestion. Traditionally, the preferred treatment modality is surgical resection. All patients in our study were evaluated by the surgeon as unsuitable for resection or had refused surgery. Further studies with larger samples sizes, a control group between TAE and surgery will help improve quality of evidence. And there are no obvious requirements of anatomical variants for TAE.

  1. Please mention numerically the loss to follow-up

Answer: Thank you very much for your suggestion. We have modified this point in the Discussion.

  1. Conclusion should be rephrased: Furthermore the medium and long-term clinical success rates demonstrate efficacy and safety of this interventional approach. (I do not think that longer follow-up would change anything. This statement is not supported by your data)

Answer: Thank you very much for your suggestion. We have modified this point in the Conclusion. I agree with your point as the reduce of hemangiomas almost within 6-12 months, hardly shrink in the last years in our study.

Reviewer 3 Report

I read with great attention the paper entitled "Medium- and Long-Term Outcome of Superselective Transcatheter Arterial Embolization with Bleomycin for Giant Hepatic Hemangioma: Results in 241 Patient"

I recognize the importance and interest of the topic addressed by the authors, however the paer is really too difficult to read (perhaps due to the lack of an expert writer) and to follow.

All the concepts are all expressed in an unclear way

Moreover, results are completely lacking

Author Response

1. I recognize the importance and interest of the topic addressed by the authors, however the paer is really too difficult to read (perhaps due to the lack of an expert writer) and to follow.

Answer: Thank you very much for your suggestion. We have modified this point and English language editing has been modified by Editage.

2. All the concepts are all expressed in an unclear way

Answer: Thank you very much for your suggestion. We have modified this point by rewriting the manuscript.

3. Moreover, results are completely lacking

Answer: Thank you very much for your suggestion. We have modified this point by rewriting the results.

Round 2

Reviewer 1 Report

The manuscript is much better now after proofreading. 

I would like Authors to add the median time period of follow-up (in months) as you have the limitation of the significantly high rate of loss of F-U

Author Response

Thank you very much for your recognition and suggestion. We have modified this point in the Results. The median and maximum follow-up times were 33 months and 60 months, respectively.

Reviewer 2 Report

Dear authors, I believe that now the article is suited for publication only minor  English spelling check is necessary especially in the newly added areas. 

Author Response

Thank you very much for your recognition and suggestion. English language editing has been modified again.

Reviewer 3 Report

The Authors were unable to satisfactory address my previous comments

Author Response

The Authors were unable to satisfactory address my previous comments.

Answer: Thank you very much for your suggestion. The manuscript has benefited from these insightful suggestions. We apologize for the manuscript is not better enough after proofreading, so we revised the manuscript again.

Previous comments:

I read with great attention the paper entitled "Medium- and Long-Term Outcome of Superselective Transcatheter Arterial Embolization with Bleomycin for Giant Hepatic Hemangioma: Results in 241 Patient"

Specific comments:

I recognize the importance and interest of the topic addressed by the authors, however the paper is really too difficult to read (perhaps due to the lack of an expert writer) and to follow.

Answer: Thank you very much for your suggestion. Although English language editing has been modified by Editage, minor English spelling check is still necessary. So English language editing has been modified again.

All the concepts are all expressed in an unclear way

Answer: Thank you very much for your suggestion. We have modified this point in the manuscript. Some concepts are expressed in the following paragraph.

Methods. Technical success was defined by the bleomycin-lipiodol emulsions were delivered into the hemangioma. Clinical efficacy was defined by indications on the imaging examinations that the tumors had decreased by more than 50% in maximum diameter of hemangiomas.

Statistical analysis. Qualitative variables were expressed as numbers and percentages. The medium and long-term efficacy rates of TAE were compared in terms of the means of the diameters of the tumors studied, and percentages of success.

Moreover, results are completely lacking

Answer: Thank you very much for your suggestion. We have modified this point in the results. There are several main results in our study are listed in the following paragraph.

First, no serious complications were observed in our study.

Second, TAE was successfully performed in all patients (100%). And, good responses were achieved in the reduction of hemangioma volumes after TAE with bleomycin. The hepatic hemangioma diameters before TAE were 9.5±3.1 cm. At 6, 12, 36, and 60 months after TAE, the hepatic hemangioma diameters were 3.6±1.4 cm, 3.0±1.3 cm, 3.0±1.8 cm, and 3.5±2.7 cm, respectively. Significant differences between the preoperative and the postoperative time-points evaluation indices.

Last, The medium (3 years) and long-term (5 years) clinical success rates were 85.2% and 86.5%, respectively. Furthermore, we found that the greatest decrease in hemangioma volume occurred at 6-12 months post-TAE, and almost no changes were observed during the follow-up period thereafter.